# Study on Dynamic Response Characteristics of Saturated Asphalt Pavement under Multi-Field Coupling

**DOI:** 10.3390/ma12060959

**Published:** 2019-03-22

**Authors:** Yazhen Sun, Rui Guo, Lin Gao, Jinchang Wang, Xiaochen Wang, Xuezhong Yuan

**Affiliations:** 1School of Transportation Engineering, Shenyang Jianzhu University, Shenyang 110168, China; gr329@stu.sjzu.edu.cn; 2College of Architecture Engineering, Chongqing University of Arts and Sciences, Chongqing 402160, China; gaolin32@163.com; 3Institute of Transportation Engineering, Zhejiang University, Hangzhou 310058, China; wjc501@zju.edu.cn; 4School of Science, Shenyang Jianzhu University, Shenyang 110168, China; 18640453168@163.com

**Keywords:** multi-field coupling, hydro-mechanical coupling, thermal-hydro-mechanical coupling, pore-water pressure, transverse stress, vertical stress

## Abstract

To study the dynamic response of saturated asphalt pavement under moving load and temperature load, 3-D finite element models for asphalt pavements with hydro-mechanical coupling and thermal-hydro-mechanical coupling were built based on the porous media theory and Biot theory. First, the asphalt pavement structure was considered as an ideal saturated fluid–solid biphasic porous medium. Following this, the spatial distribution and the change law of the pore-water pressure with time, the transverse stress, and the vertical displacement response of the asphalt pavement under different speeds, loading times, and temperatures were investigated. The simulation results show that both the curves of the effective stress and the pore-water pressure versus the external loads have similar patterns. The damage of the asphalt membrane is mainly caused by the cyclic effect of positive and negative pore-water pressure. Moreover, the peak value of pore-water pressure is affected by the loading rate and the loading time, and both have positive exponential effects on the pore-water pressure. In addition, the transverse stress of the upper layer pavement is deeply affected by the temperature load, which is more likely to cause as transverse crack in the pavement, resulting in the formation of temperature cracks on the road surface. The vertical stress at the middle point in the upper layer of the saturated asphalt pavement, under the action of the temperature load and the driving load, shows a single peak.

## 1. Introduction

There are many factors that cause the early damage of highway asphalt pavements, of which water damage is a main cause [1,2,3]. During rainfall, rainwater permeates the pavement structure through the road surface, leading to pore-water pressure and the vacuum negative suction pressure being generated under the repeated actions of vehicle loads. Due to the two effects, the adhesive properties of the asphalt pavement structure decreases, resulting in the peeling between the asphalt and the aggregate, and the detachment of asphalt membrane from the aggregate surface. This leads to water damage of the asphalt pavement [4,5,6]. In order to understand the change law of pore-water pressure, and to explain the water damage mechanism of the pavement, a number of experiments and theoretical experiments were recently carried out [7]. The increase of water in the pavement structure is significant during spring. Combined with poor drainage conditions, variations in the condition of the materials have been shown to be triggering factors that accentuated the effect of heavy vehicle loading on pavement response and damage. In one experiment, two experimental pavement sections were monitored for temperature and deflections, the section with the lowest structural capacity was found to be more sensitive to thaw weakening [8]. Analyzing and studying the deformation and damage of the asphalt pavement, which was under the combined action of seepage and vehicle load, was the main focus of study [9]. This study highlighted that the damage of the asphalt pavement emerges mainly under the vehicle load, but after seepage, the damage under the combined action of seepage and vehicle load would be more serious. Furthermore, the pore-water pressure has been shown to reach its peak on the pavement that is full of horizontal and cross crack. Moreover, the positive and negative circulation of pore-water pressure has a scouring effect on the asphalt mixture, which continually makes the pavement crack longer. Excessive water scour destroys the asphalt-aggregate interface and causes the asphalt to peel off from the surface of the aggregate, thus resulting in a loose asphalt mixture and reduced strength [10].

Researchers have discussed the issue of far field response after deriving the dynamic solution for the saturated full space [11]. All problems of saturated elastic half space can be studied by means of transformation and other methods. A similar analysis method was used to study the stratification of the elastic soil layer systematically. However, the authors of this study deemed the process to be overly complex [12]. The general expression of the time-domain solution for transient dynamic problems of multi-layered saturated soils was derived by using Laplace–Hankel joint transformation, the initial parameter method, and the transfer matrix technology [13]. The vertical vibration of elastic circular plates on porous saturated half space was studied. A further study, which considered a dynamic response model for the pavement under moving loads, was carried out with the help of a computer [14]. In another study, the dynamic response of the inner skeleton of the soil layer in subgrade soil was coupled with the permeation and consolidation of the internal water [15].

Currently, the mechanism of the coupling of the hydro-mechanical with the permeable asphalt pavement is not known. Studies on the dynamic response mechanism of permeable asphalt pavement under the combined effects of temperature, saturation, and load have not been carried out. These issues directly affect the hydraulic coupling mechanism and the dynamic response characteristics of the asphalt pavement. A large number of research has shown that the Biot consolidation theory can effectively explain the water damage of the pavement [16,17,18,19,20,21]. Therefore, based on the Biot consolidation theory, with the assumption that the pavement material is an elastic porous medium, 3-D finite element models for the asphalt pavement structures were established in this paper. The dynamic response mechanism of the permeable asphalt pavement under the coupling of hydro-mechanical and thermal-hydro-mechanical was studied. The stress, displacement, and pore-water pressure of each structure layer was obtained by a finite element simulation. By analyzing the influence of various parameters, the water damage mechanism was revealed, which provides the theoretical basis and the technical guidance for the design of a permeable asphalt pavement.

## 2. Basic Theoretical Equation

The mechanical behavior of the asphalt mixture is viscoelastic, plastic, and viscoelastic. In order to simplify the calculation and equations, the following assumptions are made:

(1) Apart from permeability, asphalt concrete is a homogeneous and fully saturated ideal elastic material.

(2) All layers of asphalt pavement are compacted uniformly, and the average porosity of all parts is equal. The porosity of all layers of the asphalt pavement is regarded as uniformly distributed.

(3) The asphalt concrete and the pore-water are incompressible, i.e., there is only deformation and no volume change.

Based on the above assumptions, when all the pores between the skeleton composed of asphalt mixture are filled with water, the pavement pores are in the state of water saturation, regardless of the existence of unsaturated pores.

### 2.1. Dynamic Equation

Based on the basic assumption of layered elastic system for the pavement structure and elastic dynamics theory, the control equation of dynamic response of pavement structure is [22]
(1)$$\left[M\right]\left\{\ddot{u}\right\}+\left[C\right]\left\{\dot{u}\right\}+\left[K\right]\left\{u\right\}=\left\{F\left(t\right)\right\}$$
where [M] is the mass matrix, [C] is the damping matrix, [K] is the stiffness matrix, $\left\{\ddot{u}\right\}$ is the acceleration of joint, $\left\{\dot{u}\right\}$ is the node speed, {u} is the node displacement vector, and {F(t)} is the load vector acting on the node.

Since the pavement structure system is a small damping structure, Rayleigh damping assumption [23] is usually adopted to express the damping matrix as a linear combination of the mass matrix and the stiffness matrix. It was is written as
(2){C}=α{M}+β{K}
where *α* and *β* are the damping coefficients related to the natural frequency and the damping ratio of the structure.

### 2.2. Control Equation for Ideal Saturated Medium

The saturated state of asphalt pavement structure is described by porous media theory, and the Biot theory [24]. In the context of these theories, the asphalt pavement structure is considered as an ideal saturated fluid–solid biphasic porous medium. The solid phase is the skeleton of the asphalt mixture, and the liquid phase is the water that fills the pores of the skeleton. The structure of the asphalt pavement is considered as a linear elastic layered axisymmetric system of homogeneity, isotropy, and in complete continuous contact between layers, and their equilibrium differential equations are established respectively. The continuity equation between the liquid phase and the solid phase is established by the mass conservation [25].

The ideal control equations for the axisymmetric semi-space of saturated porous media are
(3)$$G\left[\nabla^{2}u\left(r,z,t\right)-\frac{u\left(r,z,t\right)}{r^{2}}\right]+\left(\lambda +G\right)\frac{\partial e\left(r,z,t\right)}{\partial r}-\frac{\partial \sigma_{f}}{\partial r}=\rho \frac{\partial^{2}u\left(r,z,t\right)}{\partial t^{2}}$$
(4)$$G\nabla^{2}w\left(r,z,t\right)+\left(\lambda +G\right)\frac{\partial e\left(r,z,t\right)}{\partial z}-\frac{\partial \sigma_{f}\left(r,z,t\right)}{\partial z}=\rho \frac{\partial^{2}w\left(r,z,t\right)}{\partial t^{2}}$$

The fluid balance equations are
(5)$$\frac{\rho_{f}g}{k}\frac{\partial w_{f}\left(r,z,t\right)}{\partial t}+\rho_{f}\frac{\partial^{2}u\left(r,z,t\right)}{\partial t^{2}}=-\frac{\partial \sigma_{f}\left(r,z,t\right)}{\partial r}$$
(6)$$\frac{\rho_{f}g}{k}\frac{\partial w_{f}\left(r,z,t\right)}{\partial t}+\rho_{f}\frac{\partial^{2}w\left(r,z,t\right)}{\partial t^{2}}=-\frac{\partial \sigma_{f}\left(r,z,t\right)}{\partial z}$$

The continuous seepage equation is
(7)$$\frac{\partial }{\partial t}\left[\frac{\partial u_{f}\left(r,z,t\right)}{\partial r}+\frac{u_{f}\left(r,z,t\right)}{r}+\frac{\partial w\left(r,z,t\right)}{\partial z}\right]+\frac{\partial e\left(r,z,t\right)}{\partial t}=0$$

In these equations, *λ* and *G* are Lame constants for saturated mixture skeleton, *σ*_f_ is the pore-water pressure, *ρ*_s_, and *ρ*_f_ are the density of solid and fluid, respectively. *ρ* is saturation density of mixture, *ρ* = (1-*n*)*ρ*_s_ + *nρ*_f_, *n* is the porosity and *k* is the permeability coefficient.

## 3. Finite Element Models for Asphalt Pavements under Hydro-Mechanical Coupling

### 3.1. Finite Element Modeling and Parameter Selection

3-D finite element models for the asphalt pavements under hydro-mechanical coupling were built by COMSOL, COMSOL is a multi-physics field simulation software. The structure mechanics module and subsurface flow module in COMSOL provide a convenient was to process the calculations. In this paper, the pressure was assumed to be evenly distributed on the pavement surface. The vehicle was simplified as a uniform circular load, and the internal pressure of the tire was regarded as acting on the pavement. The equivalent circle diameter of the load was calculated as
(8)$$d=\sqrt{\frac{4P}{p\pi }}$$
where *P* is the load on the wheel, in kN and *p* is the tire contact pressure, in kPa.

The specifications of asphalt pavement design in China is *P*_max_ = 0.7 MPa. The curve of loading *P* versus time is shown in Figure 1 and is calculated by Equation (9).
(9)$$\begin{array}{l} p=P_{\max}\sin\left(\frac{\pi }{T}t\right)\;\;0\leq t\leq t_{1}\\ p=0\;\;\;\;\;\;\;\;\;\;\;\;\;\;\;\;\;\;t>t_{1} \end{array}$$
where *t*_1_ is the acting time of a single vehicle. When the speed is 40 km/h, the equivalent *t*_1_ is about 0.0225 s.

In the specifications of highway asphalt pavement design in China, 100 kN of uniaxial load is taken as the standard axial load of a double wheel group, and the wheel load is simplified as a circular uniform load. The wheel pressure is 0.7 MPa, the diameter of a single equivalent circle is *d* = 21.3 cm, and the distance between the center of the equivalent circle is *d*_1_ = 31.75 cm [26]. By Equation (9), the equivalent circle diameter of the load is calculated to be 21.32 cm. The maximum value of the instantaneous moving load *P*_max_ is taken as 0.7 MPa. When the vehicle speed is 40 km/h, the load time on an element is about 0.542 s. In order to simulate the dynamic response of the asphalt pavement under the driving load, the typical layer structure of the saturated asphalt pavement was chosen in our study. Generally, there are three layers in asphalt pavements: The surface layer contains the upper layer, the middle layer, and the bottom layer, and the thicknesses of the sublayers used in our experiment were 4 cm, 6 cm, and 8 cm, respectively; the base contained 5% cement stabilized macadam and the subbase contained 3% cement stabilized macadam, with a thicknesses of 30 cm and 20 cm, respectively. The thicknesses of the embankment was 210 cm. Taking the driving direction as the longitudinal, the direction of the load and the driving direction is perpendicular to the transverse direction. Because the pressure range of the wheel is much smaller than that of the subgrade specifications, a rectangular shape was taken as the calculation area, its transverse and longitudinal dimensions were 8 m and 6 m, respectively. By using the Mapped mapping grid, the top surface was partitioned by quadrilateral meshes. As shown in Figure 2, the total number of structural units was 13,600, and the quality of element generation was 1.0.

In this model, the position of the moving load was located at the lateral center position and the load moved from one end to the other end along a longitudinal direction. The bottom boundary of the model was in full constraint, the side boundary was constrained in the normal direction, and the upper boundary was free. Because the model was a fully permeable pavement, the lateral and top surfaces of the pavement structure all had permeable boundaries, and the initial pore-water pressure of each layer was assumed to be 0. In the porous media of the saturated bitumen, the solid phase was a skeleton structure of the asphalt mixture. The material parameters of the pavement structure are listed in Table 1 [27].

### 3.2. Analysis Results of Hydro-Mechanical Coupling from the 3-D Finite Element Simulation

#### 3.2.1. Influence of Vehicle Speed and Load Duration on Pore-Water Pressure

The material parameters and the moving loads are the two main factors that affect the hydrodynamic pressure. The change of the pore-water pressure and the transverse stress were studied by varying the loading time and the speed. The peak value of pore-water pressure under different driving speed and load duration are listed in Table 2.

Taking the logarithm of the peak value of the pore-water pressure (*p*_max_), the driving speed (*v*), and the load duration (*t*_0_), respectively, the data in Table 2 was analyzed and fitted, and the fitting result can be expressed as
(10)$$Y=3.834+0.7019V-0.2153T-0.1885V^{2}-0.2893VT-0.1455T^{2}$$
where *Y* = ln*p*_max_, *V* = ln*v*, *T* = ln*t*_0_.

In order to observe the change of the pore-water pressure (versus the speed of the saturated asphalt pavement), the midpoint of the upper layer middle plane junction was chosen as the object of study, and the time history change curve is shown in Figure 3.

As shown in Figure 3, there was a rapid increase in the pore-water pressure and a rapid dissipation the followed. The positive and negative pressures induced stripping of the asphalt from aggregate and created the cracks inside the asphalt pavement, which resulted in the premature failure of the asphalt pavement. The positive pore-water pressure also increased with the driving speed. As the driving speed increased from 40 km/h to 120 km/h, the maximum positive pore-water pressure increased from 90 kPa to 112 kPa, while the negative pore-water pressure peak changed little. Because the time of pore-water drainage decreased with the decrease of the load time, the flow time of pore-water decreased. In other words, the pore-water was subjected to a greater external pressure. Even if we disregarded the influence of pore-water flow time on the asphalt mixture, high speed driving was shown to be harmful to the asphalt mixture from the point of the pore-water pressure. By embedding grating sensors on the test road and conducting dynamic water scour tests on Marshall specimens, researchers have proved that the faster the speed, the greater the pore-water pressure, and the more serious the damage of dynamic water scour to the pavement structure [28,29,30]. As shown in Figure 4, the pressure gradient and the flow velocity directly under the load were the maximum, and the pore-water pressure increased slightly at the border because of the undrained limits.

#### 3.2.2. Influence of Speed on Transverse Stress of Structure

In the analysis of the mechanical responses of the asphalt pavement structure, the horizontal stress was observed as an important guiding factor for cracks in the asphalt pavement. When a vehicle passed, a large tensile stress was produced by the pressure acting on the road. If the stress exceeded its maximum resistance strength, cracks appeared on the surface. The transverse stress in this paper was taken as perpendicular to the direction of the vehicle and the direction of the load. To observe the transverse stress of the saturated asphalt pavement with the change of speed, the middle point of the junction of the upper layer and mid-surface was selected as the object of study, and the time history curve is shown in Figure 5.

As shown in Figure 5, the transverse stress also increased with the speed. As the driving speed increased from 40 km/h to 120 km/h, the maximum lateral tensile stress increased from 155 kPa to 190 kPa. When the vehicle passed, the load was more likely to induce cracks in the pavement. Consequently, this reduced the strength of the material and the service life of the pavement.

#### 3.2.3. Influence of Speed on Vertical Displacement of Structure

To observe the vertical displacement of the saturated asphalt pavement with the change of speed, the midpoint of the upper layer middle plane junction was chosen as the object of study, and the time history change curves are shown in Figure 6.

It can be seen that, in the case of dynamic load, the vertical displacement at the middle point of the interface of the top layer and the middle-surface layer also showed floating characteristics. When the load moved away, the maximum vertical displacement decreased. As the speed increased from 40 km/h to 120 km/h, the vertical displacement decreased from 86 (0.01 mm) to 77 (0.01 mm).

## 4. Finite Element Model of Asphalt Pavement under Thermal-Hydro-Mechanical Coupling

Based on the basic assumptions above, temperature loads were applied to the geometric models. The dynamic response of the saturated asphalt structure under the action of temperature load and traffic load was calculated.

### 4.1. Pavement Temperature

Changes in temperature and solar radiation can have an immediate impact on the temperature of the road surface and the upper layer of the pavement structure. As such, the temperature changes of the road surface should be viewed as synchronous with the temperature. Combined with the measured asphalt pavement temperature data and climatic conditions, the least square method was used to carry out the multiple regression of the road surface temperature (*T*), the average temperature of 1 h before (*T*_a1_), and the average solar radiation intensity of 3 h before (*Q*_3_). The forecasting equation of the asphalt pavement surface temperature can be described as [31]
(11)$$T=0.511+0.007T_{a1}^{2}+0.871T_{a1}+15.116Q_{3}$$
where *T* is the temperature of the asphalt pavement surface, *T*_a1_ is the average temperature of 1 h before, *Q*_3_ is the average solar radiation intensity of 3 h before.

Due to the overlay of asphalt layer on the surface, the change of temperature had little impact on the base layer and the soil foundation. Therefore, the material properties of the asphalt layer were taken as constants, and only the material properties of the asphalt layer that changed with changes in temperature were considered.

### 4.2. Pavement Structure Parameters

In the above calculations of the asphalt pavement structure, the material characteristics of the structure change with temperature were not taken into account. In fact, the material of the asphalt pavement structure was shown to be temperature sensitive, and neglecting the effects of temperature on material parameters will limit our understanding of the dynamic response characteristics of the structure. Therefore, the effects of temperature on the material parameters of the asphalt pavement need to be considered.

(1) Elastic Modulus

The elastic modulus reflects the ability of structural materials to resist deformation in the elastic stage. Generally speaking, for asphalt material, the elastic modulus decreases with the increase of temperature, and the variation is large. Therefore, the effect of temperature on the elastic modulus cannot be ignored. The relation between the elastic modulus of the material of the asphalt pavement and the temperature is [32]
(12)$$E_{1}\left(T\right)=a10^{-bT}\left(\mathrm{MPa}\right)$$
where *a* and *b* are parameters determined by experiments [33]. After fitting the data, Equation (12) becomes
(13)$$E_{1}\left(T\right)=2865.89\times 10^{-0.01984T}\left(\mathrm{MPa}\right)$$
where *E*_1_(*T*) is the elastic modulus of the material of the top layer of the pavement.

(2) Poisson’s Ratio

Poisson’s ratio is greatly influenced by temperature. The relation between Poisson’s ratio and the temperature of the asphalt mixture has been obtained through tests and data analysis [34]
(14)$$\mu_{1}\left(T\right)=0.07+\frac{1.96\times 10^{0.05T}}{117.49+10^{0.05T}}$$
where *μ*_1_(*T*) is Poisson’s ratio of the asphalt mixture and *T* is the temperature.

In the meshing of the finite element model, the heat conduction analysis module was introduced. The initial temperature of the structure was set at 30 °C. The temperature load was applied at the top layer, and it was assumed that the bottom and the four sides were insulated boundaries. A coupling stress analysis of the thermal field and the structural field was carried out.

### 4.3. Analysis Results of Thermal-Hydro-Mechanical Coupling from the 3-D Finite Element Simulation

#### 4.3.1. Spatial Vertical Stress Distributions at Different Times

When analyzing the mechanical response of the asphalt pavement structure, vertical stress is usually used as a mechanical index. Based on the built model in this paper, the vertical stress distribution law of the saturated asphalt pavement structure under the combined action of temperature and traffic load on the upper layer of pavement was simulated, and the results are shown in Figure 7.

Figure 7 shows that, with the change of temperature load, the vertical stress of the pavement structure only showed a relatively obvious change in the upper layer. However, the vertical stress of each base layer and subgrade below the surface layer also all underwent a small change. With the movement of the driving load, the application concentration area of vertical stress also moved, and the influence scope was mainly within the action scope of the driving load, while in most other areas, the stress distribution was relatively uniform, and the maximum vertical stress was about 3 MPa. To have a better view of the change of vertical stress of the asphalt pavement section, the vertical stress at different moments was taken along the *z*-axis (*z*-axis is along the road depth), as shown in Figure 8.

As can be seen from Figure 8, the vertical stress of the upper layer decreased rapidly at first, then rose slowly along the depth, and finally approached a fixed value. At the top of the pavement, the vertical stress at different times was about 2.4 MPa at the maximum, and, at the depth of about 0.005 m, the vertical stress reached the minimum. With increasing depth, the vertical stress slowly rose again to 0. The maximum vertical stress difference at different moments was about 1 MPa.

In order to better analyze the vertical stress variation, the middle point of the upper layer was taken for analysis, as shown in Figure 9.

It can be seen from Figure 9 that under the action of the temperature load and the traffic load, the vertical stress curve for the middle point of the asphalt pavement layer showed a single peak. At *t* = 0.3 s, the maximum positive vertical stress reached the maximum 2.4 MPa. The vertical stress did not show a periodic change, and it was less affected by the temperature load. Therefore, the stress was mainly caused by the driving load. However, due to the temperature load, the vertical stress curve was no longer smooth. The asphalt pavement endured the vertical action of temperature load and vehicle load. Due to the slow dissipation of the vehicle load, the load exceeded the allowable maximum load (termed as ‘shakedown limit’), resulting in rutting. For the flexible double-deck pavement system, rutting can be reduced by increasing the thickness of the first layer or changing the *E*_1_/*E*_2_ of the two layers [35]. Geosynthetic materials are widely used in conjunction with unbound materials to enhance the performance of flexible and rigid pavements. The inclusion of geo-jute on flexible pavements significantly improves the pavement performance by producing lower stress, strain, and displacement at the top of the subgrade. The improvement in reduction in stress, strain, and displacement due to geo-jute is more significant when higher wheel pressure is applied [36]. The above methods can effectively solve the rutting problem seen in asphalt pavements.

#### 4.3.2. Spatial Transverse Stress Distributions at Different Times

Due to the covering of the surface asphalt layer, the temperature change had a small impact on the base and the soil foundation. Therefore, the transverse stress of the surface layer was analyzed. To better show the transverse stress change of the asphalt pavement section, the transverse stress change curves at different times were taken along the *z*-axis (*z*-axis is along the road depth, *m*), as shown in Figure 10.

As shown in Figure 10, with the increase of time, the transverse stress on the upper layer was larger, the distribution was more uniform, and the transverse stress of the structural layers below the surface layer were all smaller.

The transverse stress of the above pavement section first decreased rapidly along the depth, then rose slowly, and finally came to a fixed value. At the top of the pavement, the maximum transverse stress was about 5.5 MPa. At a depth of about 0.008 m, the transverse stress reached the minimum value. As the depth continued to increase, the transverse stress slowly returned to 0. The maximum transverse stress difference was about 3.5 MPa. There was a great change of the transverse stress. This showed that the asphalt pavement surface was more prone to transverse tensile failure, which resulted in transverse cracks.

To better analyze the change of transverse stress, the middle point of the top layer was taken for analysis, as shown in Figure 11.

Under the combined action of the temperature load and the traffic load, the transverse stress at the middle point of the asphalt pavement layer showed a single peak. As the wheel passed the calculation point, the vertical stress of the asphalt pavement increased and the transverse stress decreased. At *t* = 0.25 s, the transverse stress visibly decreased and reached the minimum value of about 4.7 MPa. For the most part, the transverse stress changed little.

## 5. Conclusions

3-D finite element models for asphalt pavements with hydro-mechanical coupling were built based on the porous media theory and Biot theory. The spatial distribution and the change law with time of the pore-water pressure, the transverse stress, and the vertical displacement of the asphalt pavement were investigated. Following this, the asphalt pavement parameters affected by temperature were introduced into a saturated asphalt pavement finite element model, the transverse stress and vertical stress of surface layer were analyzed, and the following conclusions can be drawn:

(1) Under hydro-mechanical coupling, the pore-water pressure of the permeable asphalt pavement fluctuates periodically with the fluctuation of the external load. The fluctuation of pore-water pressure is also an important reason for the stripping of the asphalt pavement.

(2) Under hydro-mechanical coupling, the strain response of the pavement structure has the characteristic of fluctuation. At the same time, the stress response of the asphalt pavement within a certain area exhibits obvious alternating characteristics. Furthermore, there is a reverse change of transverse and longitudinal tensile and compressive stress. This alternate change is the main reason of the fatigue failure of the asphalt pavement. Under the coupling of water and force, the driving speed and the wheel load time are the two main factors that influence the peak value of the pore-water pressure. Both the speed and the load time are in direct proportion to the peak value of pore-water pressure.

(3) The dynamic response of the permeable asphalt pavement under the hydro-mechanical coupling is influenced by the speed of the vehicle. Under certain conditions, the reversal of tension and pressure occurs. As the load speed increases, the stress response of the pavement structure decreases gradually, so it is necessary to strictly control the speed in strain measurement.

(4) When a permeable asphalt pavement is in thermal-hydro-mechanical coupling, the vertical stress at the middle point of the top layer of the asphalt pavement shows a single peak. At *t* = 0.3 s, the maximum positive vertical stress reaches 2.4 MPa. Unlike the instance where the temperature effect was not considered, the vertical stress has no periodic variation. However, the variation curve of the vertical stress is no longer smooth because of the temperature load.

(5) As compared with the results of vertical stress, the temperature load has a greater influence on the transverse stress of the pavement structure. Therefore, it is more likely that the temperature load leads to the transverse fracture and the failure of the pavement.

## Figures and Tables

**Figure 1 materials-12-00959-f001:**
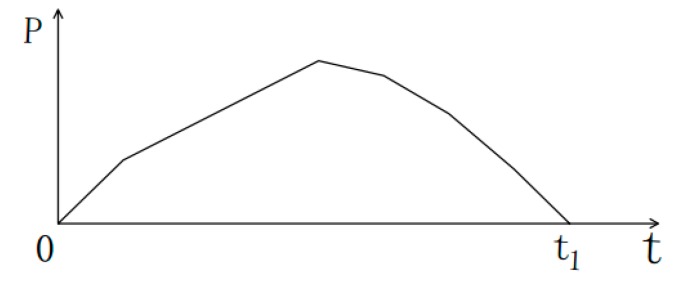
Loading curve of wheel load.

**Figure 2 materials-12-00959-f002:**
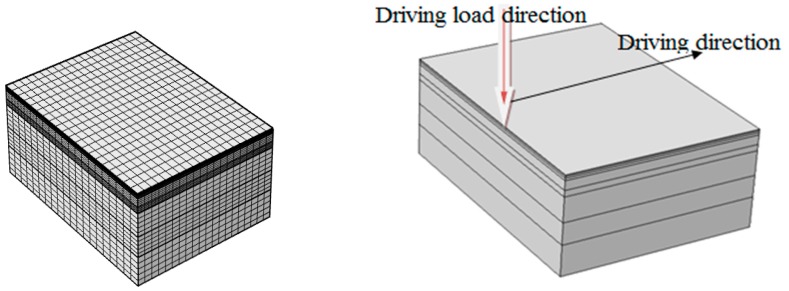
Grid and driving load direction.

**Figure 3 materials-12-00959-f003:**
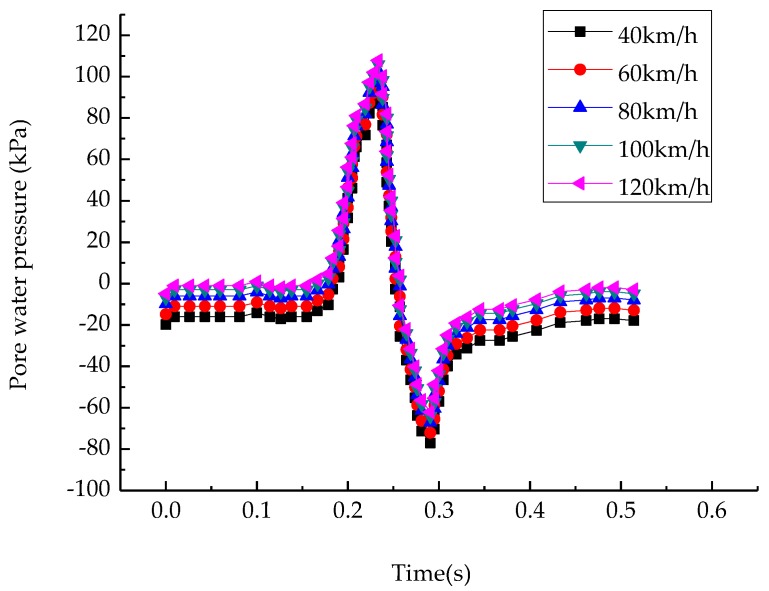
Pore-water pressure–time curves at middle points of upper-middle layers at different speeds.

**Figure 4 materials-12-00959-f004:**
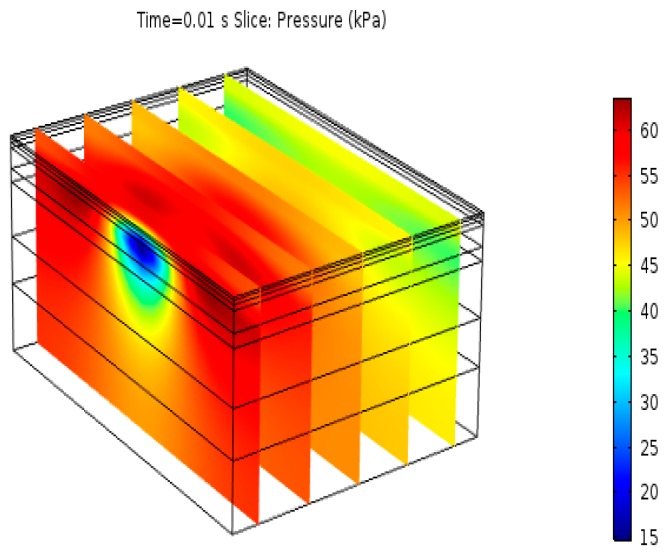
Spatial pore-water pressure distribution at 0.01 s at 40 km/h.

**Figure 5 materials-12-00959-f005:**
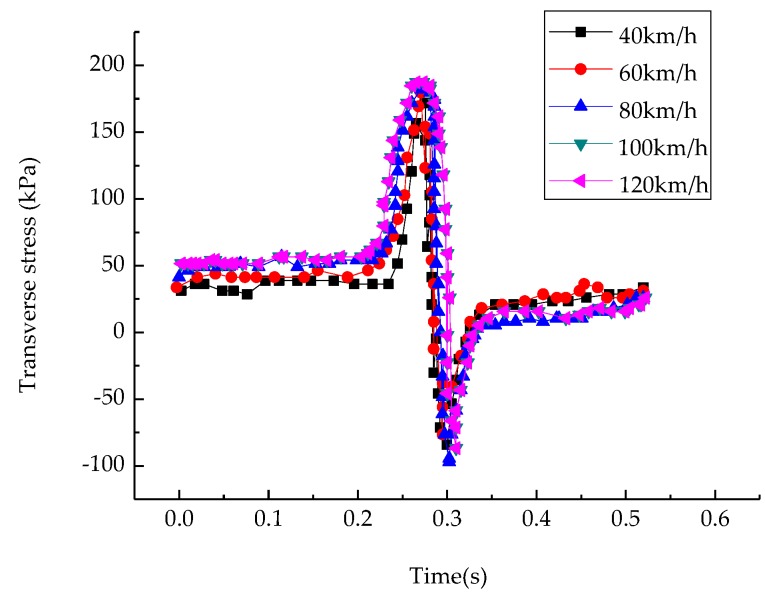
Transverse stress–time curves at middle points of upper–middle layers at different speeds.

**Figure 6 materials-12-00959-f006:**
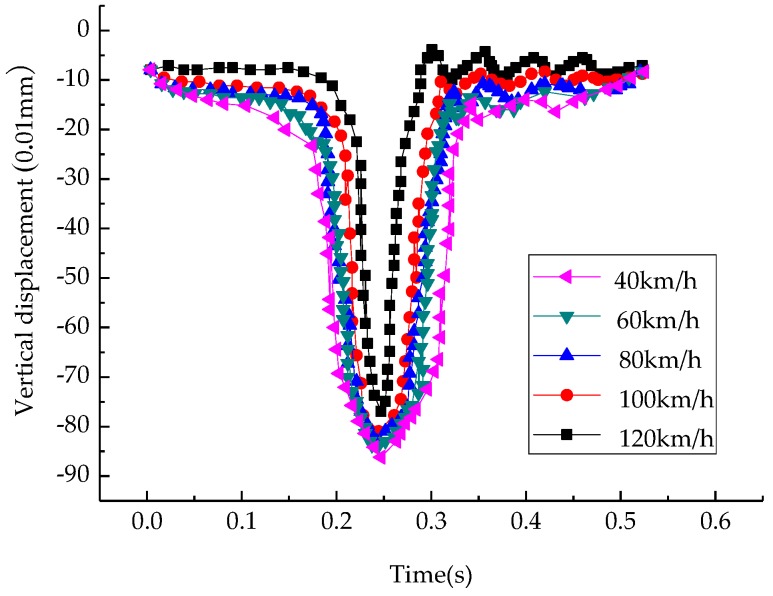
Vertical displacement-time curves at middle points of top–middle layers at different speeds.

**Figure 7 materials-12-00959-f007:**
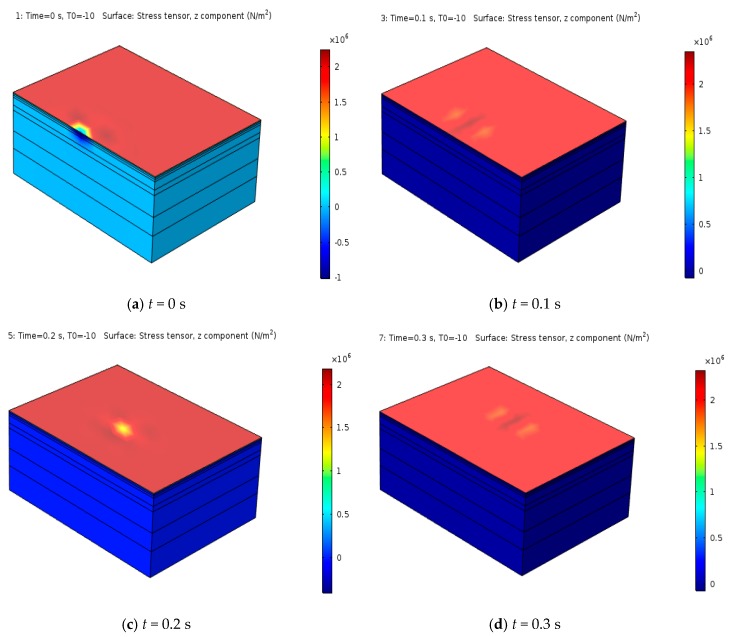
Spatial vertical stress distributions over time.

**Figure 8 materials-12-00959-f008:**
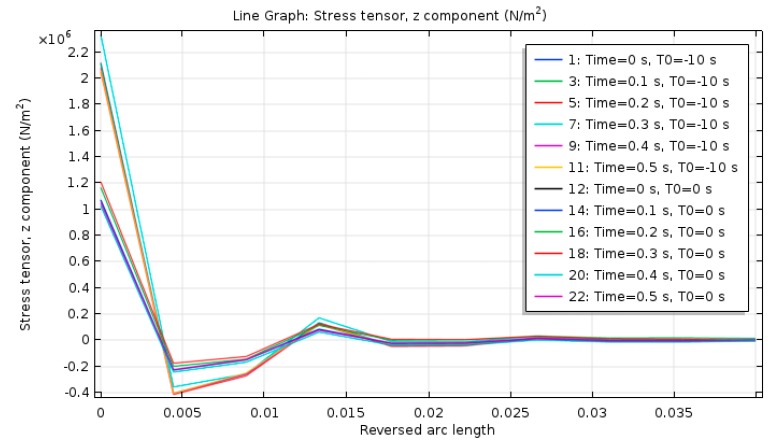
Spatial vertical stress distributions of cross section of top layer pavement at different times.

**Figure 9 materials-12-00959-f009:**
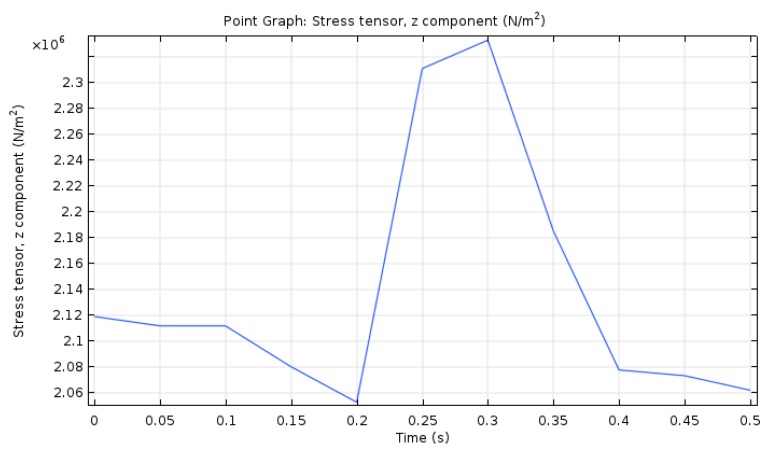
Vertical stress–time curve of middle point in top layer.

**Figure 10 materials-12-00959-f010:**
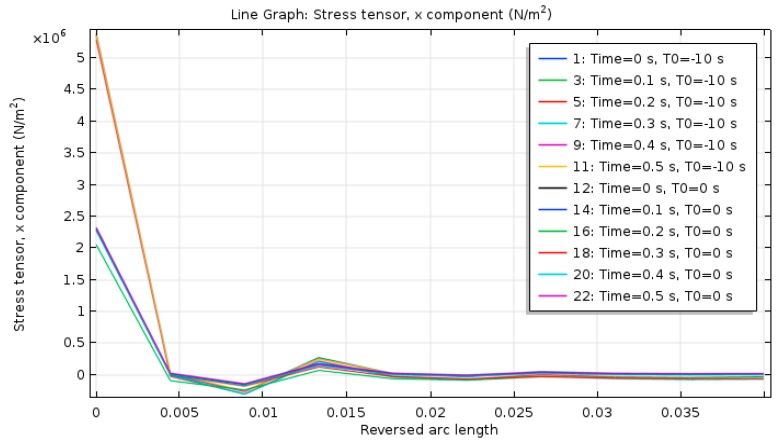
Spatial transverse stress distributions of cross section of top layer pavement at different times.

**Figure 11 materials-12-00959-f011:**
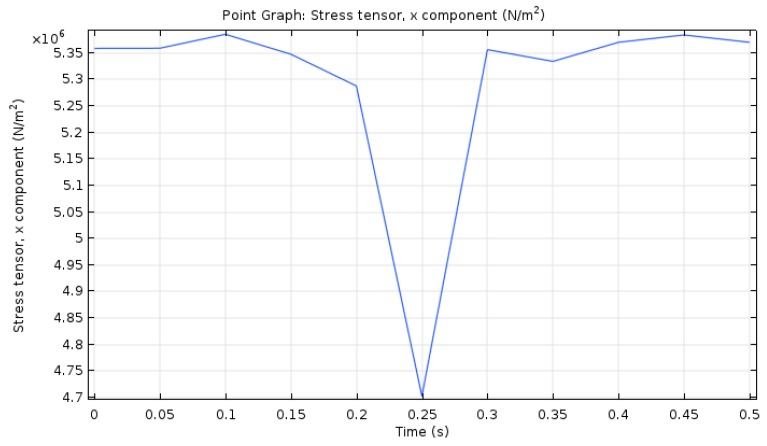
Transverse stress–time curve for middle point of top layer.

**Table 1 materials-12-00959-t001:** Structure and material parameters of asphalt pavement.

Structural Layer	Elastic Modulus (MPa)	Density (kg/m^3^)	Poisson’s Ratio	Permeability Coefficient (cm/s)
Upper layer	1400	2400	0.30	1.0 × 10^−4^
Mid-surface	1400	2400	0.35	1.0 × 10^−5^
Lower layer	1400	2400	0.35	1.0 × 10^−5^
Base	1600	2100	0.20	1.0 × 10^−4^
Subbase	600	1900	0.30	1.0 × 10^−6^
Embankment	50	1900	0.40	1.0 × 10^−6^

**Table 2 materials-12-00959-t002:** Peak value of pore-water pressure under different driving speed and load duration combination.

Load Duration *t*_0_ (s)	Driving Speed (km/h)				
	40	60	80	100	120
0.01	927.81	977.79	1027.77	1057.76	1117.75
0.05	732.97	792.01	801.66	867.36	840.64
0.1	262.57	274.76	292.91	303.58	307.16
1	51.957	42.04	32.89	15.87	17.24

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
