# Peer review of "Study on Dynamic Response Characteristics of Saturated Asphalt Pavement under Multi-Field Coupling"

_materials, 2019, doi:10.3390/ma12060959_

Round 1
Reviewer 1 Report
The manuscript named “Study on Dynamic Response Characteristics of Saturated Asphalt Pavement under Multi-field Coupling” investigated the pore-water pressure in pavement under dynamic loads. The effects of the moisture in the asphalt pavements were always of interest and this reviewer believes that the authors did a comprehensive study and considered various critical responses of asphalt. Written English is well and the manuscript reads smoothly. There are a couple of notes that the authors may apply in order to improve the quality of this research work as follows.
“Asphalt pavement structure was considered as an ideal saturated fluid-solid biphasic porous medium then the spatial distribution and the change law with time of the pore-water pressure, the transverse stress and the vertical displacement response of the asphalt pavement under different speeds, loading times and temperatures were investigated.“ This sentence is too long. Break it down to multiple sentences.
“It was is written as”….: rewrite
It should be notified that the behavior of asphalt concrete is assumed to be elastic which might not be correct especially at high temperatures. The authors might aim to simplify the equations, though, which is acceptable. That being said, it should be mentioned that the used equations do not work for viscoelastic, plastic and viscoelastic behavior.
Figures 5 and 6: The axis titles should be changed to English.
Figure 7. The notation above each schematic presentation of the stress is not clear.
Please double check the references. Some of them repeated the publication year twice.
Author Response
Thank you very much for your comments on the manuscript. We really appreciate it.

Reviewer 2 Report
This paper deals with an interesting topic. Following corrections must be addressed to improve the quality of the paper.
Comment 1: Multi-reference should be avoided. If use multi-reference, please use some significant findings from those researches. Page 1 line 35, line 40, etc.
Comment 2: Page 1, line 41, ‘a number of experiments and theoretical researches were carried out recently’, please present few examples to strengthen your statement.
Comment 3: In description of Figure 1, t is mentioned as temperature, but T is mentioned as time, but both are on same scale. Please explain.
Comment 4: Please explain why the pressure range of the wheel is much smaller than that of the subgrade specification (page 4, line 119).
Comment 5: Explain rationale to use elastic modulus of asphalt layer, those seems little lower.
Comment 6: Vehicles driving at different speed should stay different time on the pavement. Why the pore water pressure versus time were presented for 0.5 seconds for all different speeds at Figure 3? Had the point of contact been matched?
Comment 7: Longer staying vehicle should cause more damage? Did you get chance to study that? Or find otherwise, page 6 line 160: high speed driving is harmful to the asphalt mixture from the point of the pore-water pressure.
Comment 8: Figure 5 and Figure 6 axis title should be in English.
Comment 9: Page 7 line 188: As the speed increases from 40 km/h to 120 km/h, the vertical displacement increases from 0.77 mm to 0.86 mm. Could you explain this by highlighting any previous literature?
Comment 10: The author would like to cite recent publications to better describe vertical stress distribution using 3-D FEM: Rahman, M. M., Saha, S., Hamdi, A. S. A., & Alam, M. J. B. (2019). Development of 3-D Finite Element Models for Geo-Jute Reinforced Flexible Pavement. Civil Engineering Journal, 5(2), 437-446. Qian, J., Wang, Y., Wang, J., & Huang, M. (2019). The influence of traffic moving speed on shakedown limits of flexible pavements. International Journal of Pavement Engineering, 20(2), 233-244.
Author Response

(The authors gave the same response as above.)
